# Shift of Aromatic Profile in Probiotic Hemp Drink Formulations: A Metabolomic Approach

**DOI:** 10.3390/microorganisms7110509

**Published:** 2019-10-29

**Authors:** Lorenzo Nissen, Büşra Demircan, Danielle Laure Taneyo-Saa, Andrea Gianotti

**Affiliations:** 1DiSTAL―Department of Agricultural and Food Sciences, University of Bologna, V.le Fanin 44, 40127 Bologna, Italy; lorenzo.nissen@unibo.it (L.N.); saalaure@gmail.com (D.L.T.-S.); 2Food Engineering Department, Istanbul Technical University, Maslak, 34467 Sarıyer, Turkey; demircan.busra@hotmail.com

**Keywords:** *Lactobacillus plantarum*, *Lactobacillus fermentum*, *Bifidobacterium bifidum*, functional food, milk substitute, SPME-GC-MS

## Abstract

Vegetal drinks as a substitute for milk consumption are raising striking interest in the food industry. Soy and rice drinks are the most successful milk substitutes but are low in protein and fiber contents, are rich in sugars, and their cultivation systems are unsustainable; thus, alternative vegetal sources to resolve these limits must be found. A winning candidate could be hemp seed, which is a powerhouse of nutrients, is sugarless, rich in fiber and proteins, and little land and nutrients demanding. The aim is to develop novel drinks obtained from hemp seeds mixed or not with soy and rice and fermented with probiotics (*Lactobacillus fermentum*, *Lb. plantarum*, and *Bifidobacterium bifidum*). The drinks were characterized for their microbial growth, by means of culture-dependent and -independent techniques, and for their volatilome, by means of solid-phase microextraction-gas chromatography-mass spectrometry (SPME-GC-MS) analysis. The results showed that hemp seed drinks have a specific aroma and its compounds are dependent on the type of formulation and to the probiotic used. For example, in hemp seed drinks, 2-heptanol, 2-methyl, 2,4-decadienal, 2-butanone, 3-hydroxy, 2,3-butanedione, and propanoic acid were fine descriptors of probiotics fermentations. Multivariate analysis of volatile metabolites and their correlation to some physiological parameters and nutritional values offered a novel approach to assess the quality of functional hemp drinks which could result in a decisional tool for industrial applications.

## 1. Introduction

Main strategies in designing novel functional beverages may include novel functional components, enhancing the functionality and health-promoting effects (also by new technologies), development of all-natural products, and using natural food products as preservatives and sweeteners [1]. Specifically, various functional beverage products that contain probiotics have been launched, including fruits, cereals, soybeans, and vegetable-based beverages, as recently reviewed [2]. More recently, both due to the increase of lactose intolerance all over the world and the health benefits of alternative sources, a growing interest for non-dairy milk leads consumers’ demand toward novel functional beverages. Hemp seed may represent a potentially functional vegetable source for milk substitutes. Besides its nutritional advantages, it is characterized also by a low content of saturated fats and a good percentage of polyunsaturated fatty acids (PUFA) ω3 and ω6. It is naturally free of cholesterol and has a low glycemic index. Furthermore, hemp seed has striking in vitro and ex vivo antioxidant activity [3] that is lacking in soybean and rice, which are the two most common sources for vegetable drinks. Due to its high protein and magnesium content (but not calcium), hemp seed with soy and rice represents a good candidate as vegetal source comparable to macro elementary composition to cow’s milk [4]. Indeed, mixing different types of drinks could be the solution to obtain a drink formulation with the right nutritional and functional balance. In addition, the aforementioned bioactive compounds and fermentation through probiotic strains could contribute to increase the value of these beverages by classifying them as functional. On the other hand, the acidity (yogurt from cow’s milk is fermented to pH 4.6, while vegetables drinks to less than 4 [5]) and inhibitory factors (hydroxycinnamic acids, tannins) [6,7] could make vegetable drinks hard to be exploited in fermentation processes. Hemp seed drinks market is small (185 million USD) [8] due to the unpleasant taste and aspect, which limit its acceptance to the consumer. The unpleasant taste is partially derived from the biogenic amines, such as cadaverine and putrescine, which are transferred to derivate products [9], and the nonattractive aspect is because the drink is an unstable oil-in-water (O/W) emulsion system [10]. Moreover, hemp seed fermented products are missing from the market, because the vegetable source matrix is almost sugarless and microbial fermentation is hard to follow. Metabolomics has been proposed as a promising tool to assess the safety, traceability, and quality of functional beverages. Using the rapid evaluation of the metabolite profiles, it was also exploited for the determination of beverage authenticity by metabolic fingerprinting [11]. In fermented foods, the metabolite profiling was applied to observe metabolite modifications during fermentation and to predict the sensorial and nutritional quality in different fermented food matrices including dairy [12], baked goods [13], tomato [14], and few traditional fermented foods as recently reviewed [15]. This latter review summarized the increasing number of recent studies (last 15 years) on indigenous fermented beverages and foods with a special focus on the role of fermentation regarding nutritional quality, health benefits, food safety, and probiotic effects. The same authors concluded how very few papers used metabolomics as innovative approach to study traditional fermented drinks. However, as recently evidenced [16], the profiles of volatile metabolites of vegetable drinks may represent a good approach to understand the role of the type of strain as well as of the growth medium. The microbial volatile compounds perform several functions in cells and their interactions with the surrounding environment, including those related to the adaptation mechanisms. In this study, a metabolomic approach is suggested to explore different strategies to improve microbial fermentation of hemp seed drinks. Specifically, the use of hemp seed as a part of soy and rice beverage formulations together with the role of the fermentation by beneficial lactobacilli and or probiotic bifidobacteria were considered.

## 2. Materials and Methods

### 2.1. Bacterial Strains and Culture Conditions

All the bacterial strains tested belong to the microbial collection of DiSTAL (Department of Agricultural and Food Sciences), University of Bologna (Bologna, Italy) and have been previously isolated from sourdough and extensively studied [13,17,18]. *Lactobacillus plantarum* 98a, *Lb. fermentum* MR13, and *Bifidobacterium bifidum* B700795 were obtained from 30% (*v/v*) glycerol stocks stored at −80 °C and were propagated in de Man Rugosa Sharpe (MRS) (Oxoid, Thermo Scientific, Waltham, MA, USA) broth containing l-cysteine 0.05% (*v/v*) (Sigma, St. Louis, MI, USA) at 37 °C in microaerophilic conditions, applying jars with oxygen catalyst (Thermo Scientific, USA) for 48 h. 

### 2.2. Drinks Preparation 

The vegetable drinks used in this research are commercial drinks, all are organic certified products. Soy drink (S) (Alinor, Ripalta Cremasca, Italy), as stated by label, contains water, soy 8% (*v/v*), and sea salt. Rice drink (R) (Alinor, Italy), as stated by label, contains water, rice 17% (*v/v*), sunflower oil cold-pressed, and sea salt. Hemp seeds drink (H) (Bjorg, Viadana, Italy), as stated by label, contains water and hemp seeds 3% (*v*/*v*). A typical hemp milk is processed by the homogenization of ground seeds in water, and the filtered milk is thermally treated [10]. All the commercial products used in this study were UHT (Ultra-High Temperature) treated. Before fermentation, the drinks were prepared as single matrix, such as hemp seed (H), soy (S), and rice (R), or in blends. The blends were prepared as follow: 50% (*v/v*) of soy drink and 50% (*v/v*) of rice drink (SR); 50% (*v/v*) of hemp seed drink and 50% (*v/v*) of soy drink (HS); 50% (*v/v*) of hemp seed drink and 50% (*v/v*) of rice drink (HR). 

### 2.3. Fermentations

The vegetable drink samples were fermented independently by *Lb. plantarum* 98b (lp), *Lb. fermentum* PRLF (lf), and *B. bifidum* B700795 (bb), and by a bacterial consortium (c) containing equal proportion of the aforementioned strains. Cell load of inoculated bacteria was standardized at Log 6 CFU/mL (Colony Forming Unit/mL). Fermentation of the beverages was conducted in 50 mL of final volume and incubated for 24 h at 37 °C in jars with anaerobiosis catalyst (Thermo Scientific, USA). Not inoculated drinks were used as controls. Two biological replicates of each formulation were performed. At time zero, after 6 h, and at the end of fermentation (24 h), bacterial growth and pH were monitored, while volatile organic compounds (VOCs) were analyzed at the beginning and at the end of the experiment. Prior to conduct fermentation, bacterial propagation was achieved at least for two following times. Microbial load of the inocula was obtained by spectrophotometry means and measured afterward by microbial plating. Bacterial cells were centrifugated and resuspend two times in sterile water before addition to the experimental beverages for the fermentation.

### 2.4. Microbial Quantification 

Microbial quantification was obtained by both culture-dependent and culture-independent protocols. The culture-dependent protocol was achieved by plating on selective MRS agar supplemented with 0.05 g/L l-Cysteine (Sigma, USA) serial dilutions of the samples made in physiological solution (0.9% NaCl), and incubating for 24 h at 37 °C in microaerophilic conditions, using jars with oxygen catalyst (Thermo Scientific, USA). Culture-independent quantifications were obtained by qPCR with the SYBR Green I chemistry, applying genus-specific primers as Lac1 for *Lactobacillus* spp. (forward:5′-GCAGCAGTAGGGAATCTTCCA-3′ and reverse: 5′-GCATTYCACCGCTACACATG-3′) [19] and RecA for *Bifidobacterium* spp. (forward: 5′-CGTYTCBCAGCCGGAYAAC-3′ and reverse: 5′-CCARVGCRCCGGTCATC-3′) [20]. Genetic standards were prepared from relative PCR amplicons from pure cultures of the target bacterial species as described previously [21]. Extraction of bacterial DNA was obtained with Pure Link Microbiome DNA Purification Kit (Invitrogen, Thermo Scientific, USA). For both the targets, qPCR reaction on a RotorGene 6000 (Qiagen, Hilden, Germany) was set as follow: a holding stage at 98 °C for 6 min and a cycling stage made of 95 °C for 20 s and 60°C for 60 s, repeated for 45 times, followed by melting curves analysis. Quantifications were made with a five-point standard of RecA and Lac1, separately. Reactions were prepared with 1 ng of DNA, 2× Power up SYBR Green (Thermo Scientific, USA), and 250 nM of each primers (Eurofins Genomics, Ebersberg, Germany).

### 2.5. pH

The pH was determined with a pH meter (Crison, Alella, Spain) at 20 °C, appropriately calibrated with three standard buffer solutions at pH 9.21, pH 4.00, and pH 2.00. The pH values were measured in duplicate at three different times to monitor the fermentation.

### 2.6. Solid-Phase Microextraction-Gas Chromatography-Mass Spectrometry (SPME-GC-MS)

Evaluation of volatile organic compounds (VOCs) was carried out on an Agilent 7890A Gas Chromatograph (Agilent Technologies, Santa Clara, CA, USA) coupled to an Agilent Technologies 5975 mass spectrometer operating in the electron impact mode (ionization voltage of 70 eV), equipped with a Chrompack CP-Wax 52 CB capillary column (50 m length, 0.32 mm ID) (Chrompack, Middelburg, The Netherlands). The SPME-GC-MS (Solid Phase Micro-Extraction Gas Chromatography-Mass Spectrometry) protocol, while the identification of volatile compounds that was employed was previously published [13]. Before each head space sampling the fiber was exposed to the GC inlet for 10 min for thermal desorption at 250 °C in a blank sample. The samples were then equilibrated for 10 min at 50 °C. The SPME fiber was exposed to each sample for 40 min and finally the fiber was inserted into the injection port of the GC for a 10 min sample desorption. The temperature program was: 50 °C for 0 min, then programmed at 1.5 °C/min to 65 °C, and finally at 3.5 °C/min to 220 °C, which was maintained for 20 min. Injector, interface, and ion source temperatures were 250, 250, and 230 °C, respectively. Injections were carried out in splitless mode and helium (3 mL/min) was used as carrier gas. Identification of molecules was carried out by comparing their retention times with those of pure compounds (Sigma, USA) and confirmed by searching mass spectra in the available databases (NIST version 2005 and Wiley version 1996) and literature. Ethyl alcohol and acetate were absolutely quantified in mg/Kg, while all other VOCs were relatively quantified in percentage.

### 2.7. Statistical Analyses

All statistical analyses were performed using TIBCO Statistica 8.0 (Tibco Inc., Palo Alto, CA, USA). Normality was checked with the Shapiro–Wilks test and homoscedasticity was evaluated with the Levene’s test [22]. Differences between all samples were evaluated with Analysis of Variance (ANOVA), while Principal Component Analysis (PCA), K-mean clustering, Spearman Rank Correlations, and Two-way joining heatmap were used to study the relationship between the variables. For post hoc test, a Tukey’s test was employed. For PCA and Spearman Rank Correlations, the data were normalized using the mean centering method. All results are expressed as mean values obtained at least from duplicate batches in two independent experiments. 

## 3. Results

From the volatilome analysis through SPME-GC-MS, among 48 cases, more than 200 molecules were identified and approximately 120 were quantified relatively. For a landscape description of the volatilome, two options were chosen: i) related to not fermented cases, a data set of 46 significant molecules (*p* < 0.05) was generated prior to normalization with the mean centering method (Section 3.1); ii) related to all cases, a dataset with the sums of relative abundances of molecules organized by chemical classes was employed to compare the not fermented cases to the means of fermented cases (Section 3.2). Afterward, for a more specific investigation, and in order to generate robust data trainings for multivariate analysis, two other options were chosen: (i) the most abundant compounds were set apart, such as ethyl alcohol and acetic acid, and then independently quantified in mg/Kg, employing an internal standard as described previously [18] (Section 3.3); (ii) with the exclusion of those two compounds, multivariate analyses (PCA, K-Means, Spearman Rank) were obtained from four super-normalized data sets organized by different VOCs chemical classes (Section 3.4). Moreover, to originate Spearman Rank correlations, independent chemical class sorted VOCs were correlated to a normalized dataset obtained from nutritional label values, pH values, and microbial quantifications, whose raw data are supplied in Appendix A. Nutritional values were obtained from labels, and we considered scientifically satisfactory as those are regulated by Regulation (EU) No 1169/2011 of the European Parliament and of the Council [23].

### 3.1. Quantification of VOCs in Nonfermented Drinks

Among all the molecules quantified, aldehydes showed the major abundances and hexanal, heptanal, 2-heptenal (Z), octanal, and nonanal permitted to discriminate between matrices. Hexanal was present and quantified as one of the most abundant compounds mainly in every case, reaching the top level in hemp seed drink (H). Even heptanal scored the highest value for H and little less for rice drink (R), but was slight in soy drink (S) and in the blends. 2-heptenal (Z) again hit the top for H, at a lower value in hemp seed drink blends and scarcely for the other matrices. Octanal and nonanal were matrix specific for the R drink, indeed these compounds were undetected in H and two-folds less quantified in S. Nevertheless, benzaldehyde was quantified in a relatively high rate in S (Appendix A).

### 3.2. Comparison of Groups of Molecules Before and After Fermentation

During fermentation a general trend was evidenced by every matrix with different effects: alcohols and alkanes increased in abundance, while aldehydes and ketones decreased (Figure 1). Considering the class of alcohols, except ethyl alcohol, related compounds were almost undetected in not fermented (NF) drinks. After fermentation, the highest sum was scored by SR and similarly for S and R, which have approximately three-fold higher quantity than H. Ketones did not show homogeneous conduct among the cases; indeed, after fermentation, their abundance was higher for H and hemp seed-based blends, but was lower in R and S and in SR blend. Finally, for aldehydes quantitation, besides a slight reduction of abundances after fermentation, no significant differences were seen among diverse matrices. Anyhow, the decrease was more evident in single matrices than in their blends.

### 3.3. Quantifications of the Main Fermentation Metabolites: Ethyl Alcohol and Acetic Acid

Considering major molecular compounds as ethyl alcohol and acetic acid, results of quantification in mg/Kg of fermented matrices are reported in Figure 2 as means of the two replicates and two independent experiments. Ethyl alcohol in 100% soy drink (S) fermented with *Lb. fermentum* MR13 (Slf) had the highest concentration, accounting for circa 17 mg/Kg, while on the same matrix *B. bifidum* 700795 (Sbb) accounted for the lowest concentration. Rice matrix samples (R) scored a mean value of 12 mg/Kg, reaching the top for the case fermented with the consortium (Rc) and the lowest for the case fermented with *B. bifidum* 700795. Albeit hemp fermented by *Lb. plantarum* 98b (Hlp) recorded the minor average abundance of only 3 mg/Kg, the uppermost ethanol concentration of 100% hemp seed beverages (H) was obtained when fermented with the consortium (Hc). Interestingly, in contrast to all other results, this set of cases that fermented with *B. bifidum* 700795 was not the worst performer. In every blended matrix, quantification of ethyl alcohol was lower than those of S or R but higher than that of H. Moreover, the blends’ results showed similar trends, with concentration of ethyl alcohol ranging from 8 to 6 mg/Kg and always with *B. bifidum* 700795 as the worst performer (Figure 2A). Results from acetic acid quantifications (Figure 2B) had an opposite trend in respect to that of ethyl alcohol; indeed, the blends scored higher values than the single matrices. The mean concentration of hemp–rice blend (HR) cases reached the highest values, hitting 7.5 mg/Kg, followed by that of hemp–soy blend (HS) with 6 mg/Kg. On the contrary, low abundances of acetic acid were observed for single matrices, as the major value was recorded for R, with just 3 mg/Kg. Among HR the highest value was reached by the case fermented with *B. bifidum* 700795 (HRbb) while *Lb. plantarum* 98b scored the top value among HS. Finally, the blend samples fermented by consortium reached the maximum in the soy–rice blend (SR). Little differences among the cases were instead found on set of single matrices.

### 3.4. Multivariate Analysis of VOCs Organized by Different Chemical Classes

#### 3.4.1. Alcohols

Significant differences (ANOVA, *p* < 0.05) on the dataset of alcohols obtained from GC-MS analyses of VOCs were defined for ten different molecules. PCAs of cases and variables (Figure 3A,B) were made accordingly to K-means cluster analyses (data not shown) that permitted to shape four clusters and plot the cases bestowing to factor coordinates and their Euclidean distance from the plane center. Cluster 1 (three cases) grouped just cases from HS blend, while the fermented drink with the consortium was outcasted (HSc). It was mostly characterized by 2-heptanol, 2-methyl, and 1-octen-3-ol, and this latter compound was detected from three- to ten-fold higher than other groups. Cluster 2 (seven cases) included two R drinks, two S drinks, and three out of four of HR. This cluster was described by the top concentration in heptanol (17-fold higher than other clusters). In Cluster 3, six cases were reported: all four SR blends and two hemp seed blends both fermented with the microbial consortium (HSc and HRc). Moreover, *B. bifidum* was present in four out of six cases of Cluster 3. This cluster was described by the highest concentrations in 2-hexanol and octanol, almost 10-fold and 5-fold higher than each other, respectively. Moreover, just the cases in it are described by the presence of maltol. Cluster 4 (eight cases) comprises H drinks and all blends fermented with the microbial consortium. However, relative minor abundances of all the significant alcohols and a high abundance of cyclohexen-2-ol were the discriminant variables. By using the Spearman rank analysis, the correlation of fermentation performances (pH decrease and microbial increase) and beverage composition (nutrition facts) with VOCs were studied for the whole data set, independently on the different drink formulations (Figure 3C). Based on the PCA results, heptanol, 2-heptanol, 2-methyl, 1-octen-3-ol, and cyclohexanol were those alcohol compounds that described better the hemp-based matrices. Heptanol and 2-heptanol, 2-methyl were solid descriptors of both HR and HS. Indeed, heptanol was linked to HR and 2-heptanol, 2-methyl to HS. Heptanol can be found in different plant species and generates a characteristic pleasant aroma with a mild alcohol odor. Its presence in our samples was described by the acidification of the matrix due to fermentation, indeed the correlation with “Delta [pH t24]” was positive and significant (*p* < 0.05). For this compound, positive and significant (*p* < 0.01) correlations were found even to higher value of “Energy”, “Carbohydrates,” and “Salt”, while, on the contrary, a negative correlation was that related to fats (*p* < 0.01). So far, it is speculated that heptanol was related to intrinsic issues of hemp seed matrices, providing substrate for microbial fermentation (pH decrease) more than for their growth (no significant correlation with this latter). A generally opposite trend was that of 2-heptanol, 2-methyl, which resulted significantly (*p* < 0.01) but negatively correlated to “Energy”, “Carbohydrates”, and “Salts” and, in contrast, significantly (*p* < 0.05) and positively to variables specific of the microbial growth, such as “Delta count t24”. Indeed, similar results for this latter compound were seen in doughs fermented with the same strain of *Lb. plantarum* producing high amount of 2-heptanol, 2-methyl after fermentation [24]. Cyclohexen-2-ol described pretty well the cases including all single hemp fermented drinks and was significantly (*p* < 0.05) and negatively correlated to the variables “Fiber” and to both bacterial growth parameters (after 6 and 24 h).

#### 3.4.2. Aldehydes

With the same approach, GC-MS data of 13 different aldehydes (ANOVA *p* < 0.05) were analyzed in a PCA model for both cases and variables (Figure 4A,B). Cluster 1 collected just two cases, i.e., H and S fermented with *B. bifidum,* and it was the only one described by the presence of butanal, 3-methyl and by the highest concentration in heptanal. This latter was four-fold higher than cluster 2, and just in traces or undetected in the other cases. All the six cases of Cluster 2 included hemp matrices and four out of six of them were fermented by *B. bifidum*. This cluster among all was described by the highest quantity in nonanal and in 2-heptenal (Z) (usually scarcely or not present in the other clusters). A similar trend was seen for cluster 3 made by ten cases, heterogeneously distributed regarding matrix, while preferably fermented by lactobacilli (seven out of ten drinks). This cluster among all was described by the highest concentration in decanal, ranging from 3-fold to more than 6-fold higher. Cluster 4 was formed by five cases, heterogeneously distributed regarding the matrix variables but including four cases of *B. bifidum*. This cluster among all was described by the highest abundance in 2,4-decadienal (E,E) that was instead found in traces in the other cases. Considering Spearman rank correlations of fermented drinks, based on PCA results over VOCs of the class of aldehydes, good descriptors of the hemp seed drinks were the dependent variables of 2-heptenal (Z) and nonanal for H, decanal for both HS and HR blends and 2,4-decadienal (E,E) when these were fermented with *B. bifidum*. Considering Spearman Rank correlations (Figure 4C), 2-heptenal (Z) formation was linked to the variables of fermentation process but not directly to microbial growth, as it was significantly (*p* < 0.05) and positively correlated to variable “Fats”, while significantly (*p* < 0.01) and negatively correlated to variables “Energy” and “Carbohydrates”. Decanal matched to be significantly (*p* < 0.01) and positively correlated to microbial growth (both after 6 and 24 h of growth) and, with significance (*p* < 0.05), was positively correlated even to “Fiber” and “Proteins” variables. Thus, it seemed that both composition and microbiota increased the abundance of decanal. Lastly, 2,4-decadienal (E,E) was significantly (*p* < 0.01) and positively correlated to “Delta count t24” variable, indicating a direct influence of bacteria (although they need 24 h) in generating high abundances of this aldehyde. In particular and in line with other literature insights [25], 2,4-decadienal (E,E) from our dataset was linked to the presence of *B. bifidum*. Indeed, from PCA of aldehydes, four out of five members of the cluster described by 2,4-decadienal (E,E) were those fermented with the presence of *B. bifidum* strain.

#### 3.4.3. Ketones

Analogous to the previously described approach to GC-MS data analysis of the classes of ketones, significant differences (ANOVA, *p* < 0.05) on 14 different ketones separated three clusters plotted in Figure 5A,B. Cluster 1 (six cases) included just hemp seed formulations, three of them fermented with lactobacilli and three with the bacterial consortium. Cluster 1 cases accounted for the top concentration for 2,3-butanedione and 2-butanone, 3-hydroxy (from 10- to 20-fold more than the other clusters). Cluster 2 grouped three cases (two made with hemp–soy matrix and one with rice matrix) and two of them were fermented with the sole *B. bifidum*. Cluster 2 was described by the highest values in the dataset in 2-heptanone, acetone and 2-decanone. Cluster 3 included the majority of the cases (*n* = 15). In nine of them soy was present in different formulations including S and SR. This cluster was described by general lower relative abundances for all ketones. Similar trends were already seen for other classes of molecules when a cluster grouped mainly single matrices (nine cases in cluster 3). Nevertheless, two compounds here appeared to be specific and unique, 2-pentanone, 4-methyl, and 2-butanone (in other clusters found in traces or not detected, respectively). Interestingly, 2-butanone, 3-hydroxy and 2,3-butanedione robustly described a PCA cluster recipient of sole members of hemp-based matrices. When the data-training of these VOCs abundances went through Spearman Rank, for both of them were found significant (*p* < 0.05) and negative correlations with “[Delta pH t24]”, “Salt”, “Energy”, and “Carbohydrates” variables, while positive correlations just with “Fats” (Figure 5C). 2-butanone, 3-hydroxy is known as acetoin, while 2,3-butanedione, which is a secondary metabolite naturally occurring during beverages fermentation, confers a pleasant buttery taste to the drinks. 

Following the same aforementioned procedure, significant differences (ANOVA, *p* < 0.05) on the dataset of alkanes obtained from GC-MS analyses of VOCs were defined for 17 different molecules. PCA plots of cases and variables (Figure 6A,B) accordingly to K-means clustering defined three clusters. Cluster 1 (eight cases) grouped six formulations with rice source, all were fermented by lactobacilli (*Lb. plantarum* for five of them). This cluster was described by the highest concentration among all in decane, dodecane, and tetradecane, and these latter two were found in traces or undetected in the other two clusters. In cluster 2 (ten cases), seven drinks contained soy in their formulation, while seven contained hemp seed drinks. Moreover, both the series of S and HS were here included out casting just one case. This cluster was described to be the sole accounting for heptadecane and octadecane. The six cases of cluster 3 were all based on sinlge matrix formulations clustering three out of four of R. Albeit considering bacteria independent variables, it has to be mentioned that four cases included *B. bifidum*. This cluster was described by octane, 2,6-dimethyl and nonane, 2-methyl, which were both present approximately ten-fold more than in other clusters. Considering Spearman rank correlations on alkanes of fermented drinks, for those alkanes describing the hemp-based drinks, no significant correlation was found to microbial fermentation parameters. Otherwise, heptadecane and octadecane were good descriptors of cases related to hemp drinks, and both compounds were found to be positively correlated to “Proteins” and negatively to “Energy Kcal/g” and “Carbohydrates”, while octadecane was positively correlated even to “Fats” (Figure 6C). The cluster of PCA whose descriptors were these two molecules contained most of the cases related to hemp seed drinks but even a similar number of cases related to soy. Considering the results from Spearman rank correlations, it was possible to speculate that both heptadecane and octadecane were more linked to hemp drinks than to soy drinks, because hemp drink has a higher concentration of proteins and oppositely a lower content of carbohydrates than soy.

Following the same aforementioned procedure, significant differences (ANOVA, *p* < 0.05) on the dataset of fatty acids obtained from GC-MS analyses of VOCs were defined for nine different molecules (Figure 7A,B). In this work, we decided not to include the data-training of acetate quantifications in the dataset we used to compute PCA and Spearman Rank analyses, because for the purpose to describe the aromatic profile made by VOCs, its quantity values were overwhelming the values recorded for the other fatty acids, thus was not possible to highlight the abundance and weighs of the most of the fatty acids found by SPME-GC-MS technique. Cluster 1 included ten cases principally derived from drinks samples of S or R and/or relative blends. The other remaining cases not of hemp seed origin were contained in cluster 4. Otherwise, cluster 2 and cluster 3 were those including the hemp seed matrices. These two clusters were distant to each other and were drawn in two almost opposite sector of the PCA planes. So far, cluster 2 comprised four cases of hemp blends, such as HR and HS, and, in particular two were fermented with *Lb. fermentum* and two with *B. bifidum* and the consortium. This cluster was mainly described by a higher diversity of fatty acids, such as propanoic acid, propanoic acid, 2-methyl, butanoic acid, octanoic acid, nonanoic acid, and propanoic acid, 3-hydroxy (lactic acid). In contrast, Cluster 3 included six cases and four of these were those of H series, but this latter cluster was described by fewer species of acids, such as hexanoic acid, pentanoic acid, and butanoic acid. Considering Spearman rank correlations, the acids variables that described the hemp seed drinks, likewise lactic acid, propanoic acid, octanoic acid, and nonanoic acid, were differently correlated to the physiological variables (Figure 7C). For example, lactic acid was positively and strictly significantly correlated to the decrease in pH during fermentation (*p* <0.01), and to the increase in microbial loads (*p* < 0.05). Otherwise, it was negatively correlated to fats content (*p* < 0.05). So far, the trend of this compound was bound to microbial activity. Even propanoic acid seemed directly linked to bacterial fermentation, because it was positively and significantly (*p* < 0.01) correlated just to both the variable of microbial growth, as “delta count t6” and “delta count t24”, and no other correlations were significant. The same trend and correlation significance were remarked even for octanoic acid and nonanoic acid (*p* < 0.01). Additionally, octanoic acid was positively correlated even to fiber content (*p* < 0.01), confirming that it was a compound properly bound to hemp seed drinks that have the top proportion in fibers.

## 4. Discussion

Clustering results from PCAs of significant VOCs drew a general and robust scenario where hemp seed based matrix variables were described mainly by: (i) higher concentration of alcohols such as heptanol, when hemp was blended with rice, and 2-heptanol, 2-methyl, and 1-octen-3-ol, when the blend was made with soy; (ii) higher concentration of aldehydes such as, nonanal and 2-heptenal (Z); (iii) higher quantity of ketones, such as, 2-butanone, 3-hydroxy and 2,3-butanedione; (iv) higher concentration of alkanes, such as heptadecane and octadecane; v) higher proportion of medium chain fatty acids. Spearman rank correlations additionally gave us more insights as: (i) alcohols, such as cyclohexen-2-ol and 2-heptanol, 2-methyl, were related in the hemp seed drinks to intrinsic features of the matrix and to microbial fermentation, in particular to that steered by *Lb. plantarum*, respectively; (ii) aldehydes, such as 2-heptenal (Z) and nonal were characteristic of the hemp seed drinks and were correlated mainly to the matrix, while 2,4-decadienal production was driven by microbes; (iii) ketones, such as 2-butanone, 3-hydroxy, and 2,3-butanedione, were related to microbial fermentation but received a boost in their accumulation by the hemp matrix, more than the soy or rice matrices; (iv) alkanes, such as heptadecane and octadecane, were positively correlated to high content of proteins and fats and inversely correlated to carbohydrates and energy, thus were more related to hemp seed matrix than to soy; indeed, the former is almost five times richer in protein, has an high content of unsaturated fatty acids, and is depleted in sugars; (v) acids, as medium chain fatty acids, likewise octanoic acid (caprylic acid) and nonanoic acid (pelargonic acid) are found in hemp seed and were mainly correlated to the matrix, while propanoic and lactic acid seemed correlated to the microbial inocula.

### 4.1. Alcohols

Heptanol, 2-heptanol, 2-methyl, 1-octen-3-ol, and cyclohexen-2-ol were the alcohols compounds, except ethyl alcohol, that discriminated more the hemp seed matrices. The first two were more abundant in hemp blends, while the latter was quantified just in single hemp drink. In literature, these compounds are not yet described in hemp fermented products. Although, in the food industry, these are used as flavoring agents and were reported to be present in rice and soybean raw materials or products, conferring a typical olfactory issue described as musty, pungent, leafy green, and mushroom taste, respectively [26]. Moreover, 1-octen-3-ol is reported to be a natural product derived from linoleic acid during oxidative breakdown, which can be found in different plant and mushroom species and owns antimicrobial activity against spoilage and opportunistic microbes [27]. Cyclohexen-2-ol is found in tea leaves and exhibits sweet, floral, caramelized, honey-like notes associated with high-quality tea [28]. From our analysis, these alcohols were not present prior to fermentation, thus is indicative that were produced during fermentation but independently from the inoculated strain used. 

### 4.2. Aldehydes

2-heptenal (Z), nonanal, and decanal were the aldehydes mainly characterizing the hemp seed matrices. The presence of these compounds confers a typical olfactory tract. 2-heptenal (Z) has an almond flavor and from our results seemed linked both to fat composition and fermentation process, which is in line with literature findings where the abundance of 2-heptenal (Z) increased during sourdough fermentation [29]. 2-heptenal (Z) is found in pulses and sometimes detected in soybean raw material [30] but was not found in fermented soy products [26], nor in those with rice [31]. Nonanal and decanal has green and soapy aroma characteristics [32]; the former is reported to be associated with buckwheat and rice products and has a rose-orange odor [33], while the latter is found to be associated with orange juice sensorial volatile [32]. From our results, 2-heptenal (Z) seemed to well characterize hemp matrix, compared to soy and rice, while the addition of hemp to soy and rice blends improves the abundance of decanal in the final products, conferring a specific aroma.

### 4.3. Alkanes

No particular alkanes significantly described the single hemp matrix. However, in blends with soy and rice, heptadecane (greatly increased by fermentation process) and octadecane characterized these groups of cases. Both these compounds are acyclic aliphatic alkanes that are reported severally to be plant and microbial metabolites, but scarce information is available. While heptadecane is a tasting alkane when binds to saliva and is typical of licorice, pepper, and other essential plants oils [34], it has even been described in spirulina powder [35] and found in traces in *Pyrus serotina* [36]; octadecane has a fuel-like smell, is found in alcoholic beverages with limited odor threshold [37], and is associated to hop [38]. This latter feature is in line with our findings, since hemp and hop are the only two genera of the *Cannabaceae* family; moreover, no reports were previously produced over this compound presence in every matrix tested in the current work. 

### 4.4. Ketones

2,3-butanedione and 2-butanone, 3-hydroxy produced by microbial fermentation were the ketones that better describe the hemp seed fermented matrices. Indeed, besides their correlation with fat content, the influence of microbes to produce 2-butanone, 3-hydroxy and minorly 2,3-butanedione is known [39], but so far from our results a large stimulus was derived from the matrix itself and the fermentation process. 2,3-butanedione is found associated with fermented balsamic vinegars in high concentrations [40] and is found by volatile analysis of beer [41]. The impact of 2-butanone, 3-hydroxy and 2,3-butanedione on the final products are relative to better texture and aroma. In fact, 2,3-butanedione is used to improve aroma of butter or to make beer lighter to the palate and 2-butanone, 3-hydroxy to confer a pleasant almond aroma and to improve the texture. Our results over 2-butanone, 3-hydroxy on hemp matrix variables indicated that this molecule was depending even on *Lb. plantarum* and are in line with some experimental findings [42]. 

### 4.5. Organic Acids

Caprylic and pelargonic acids are medium chain fatty acids that can be found in hemp seed, and in particular the first ranges up to 2.5% of total fatty acids in the lipid matter [43]. Both acids are detectable in traditional soy and rice fermented products and are indicated as metabolites of microbial fermentation [44], but the presence of these compounds in soy or rice not fermented matrices is unreported. From our results, prior to fermentation, caprylic acid was quantitated just in the hemp-based matrices; otherwise, pelargonic acid was quantitated in all matrices, except single hemp, thus it can be reasonable that its production was driven by microbial fermentation, which was major when soy or rice were blended with hemp. Caprylic and pelargonic acids have a particular health effects on the host, indeed are effective on excessive calorie burning, inducing weight loss [45], and binding to -OH of serine residues of ghrelin activate the hormone and regulate hunger [46]. Caprylic acid was recently used in combination to antioxidant compounds, such as trans-resveratrol, to produce esters lipophenols that have stronger and more stable host antioxidant activity [47]. Instead, pelargonic acid is found in whey-fermented alcoholic beverages [48]. In contrast, the aroma of these compounds is not very pleasant. Indeed, while pelargonic has a waxy, dirty, and cheesy with a cultured dairy nuance odor and a taste defined as fatty, waxy, and cheesy with a mild sweet creamy background [49], caprylic acid is known to have a slightly unpleasant rancid-like smell and taste [50].

## 5. Conclusions

The development of foods that has a low glycemic index, low in saturated fats, carbohydrates and additives, and lactose-free, but have high energy value derived from good fats and fibers is fundamental for the well-being of healthy people and the intolerant ones. Considering this scenario, the food sector of dairy products has found alternatives to cow’s milk with vegetable drinks. Otherwise, the most common vegetable sources exploited, soy and rice, are steering a massive market and the feedstocks look not sufficient nor sustainable to satisfy the global demand, as demonstrated by the recent intense Amazon forest burnings or by the rising custom duties. The search for alternative vegetable matrices is open and wide and a winning candidate could be hemp seed, which has a high nutritional value and generates higher yields in smaller fields. Although health and nutritional potential of hemp seed are well known, the vegetable drink derived is still not common, because is less appetible for the consumer due to less pleasant taste, aspect, and aroma. This study was conducted to improve the hemp seed drink, mixing with soy or rice drinks and fermenting with probiotic. For the first time, a metabolomic approach to study the sensorial characteristics of different hemp drink formulations is proposed. The results showed that specific aromatic compounds are dependent not solely to the type of formulation used, but even to the probiotic inoculated. The hemp seed drinks that we presented have a specific aroma characterized by high abundances of heptanol, heptenal (Z), 2,3-butandedione, octadecane, caprylic acid, and pelargonic acids. This study for the first time offers the metabolomic profiles of the aromatic VOCs of different hemp seed fermented drinks, in order to indicate possible solutions to obtain the best or desired formulations.

## Figures and Tables

**Figure 1 microorganisms-07-00509-f001:**
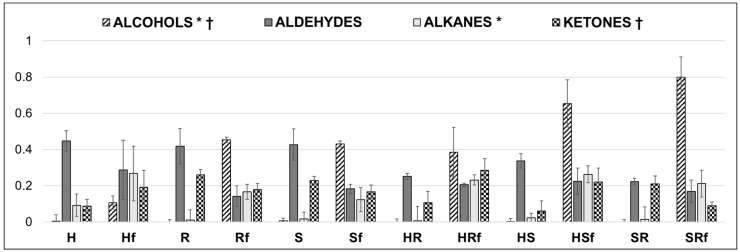
Sums of groups of molecules from not fermented and from means of fermented. *p* < 0.05; † = alcohols vs alkanes not significant; * = alcohols vs ketones not significant. H = hemp drink; Hf = fermented hemp drink; R = rice drink; Rf = fermented rice drink; S = soy drink; Sf = fermented soy drink; HR = hemp/rice drink; HRf = fermented hemp/rice drink; HS = hemp/soy drink; HSf = fermented hemp/soy drink; SR = soy/rice drink; SRf = fermented soy/rice drink. Y axis reports normalized (mean centering method) values of volatile organic compounds (VOCs) quantification.

**Figure 2 microorganisms-07-00509-f002:**
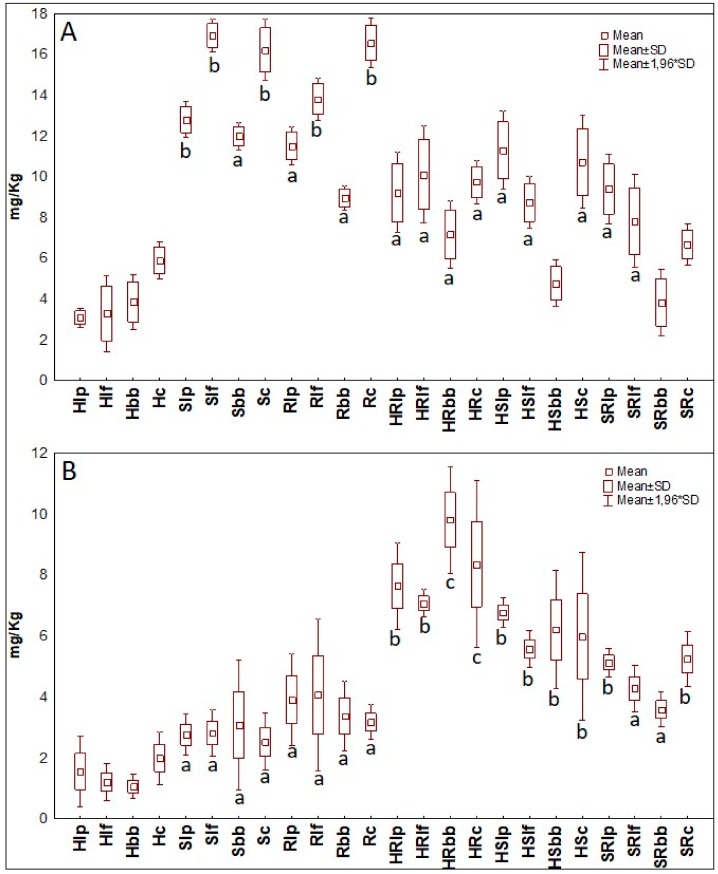
(**A**) Mean values in mg/Kg of ethyl alcohol for each fermented case. (**B**) Mean values in mg/Kg of acetic acid for each fermented case. a, b, c = Means with different letters are significantly different at *p* < 0.05 by Tukey’s test. Hlp = 100% hemp seed drink fermented with *Lb. plantarum* 98b; Hlf = 100% hemp seed drink fermented with *Lb. fermentum* MR13; Hbb = 100% hemp seed drink fermented with *B. bifidum* B700795; Hc = 100% hemp seed drink fermented with bacterial consortium; Rlp = 100% rice drink fermented with *Lb. plantarum* 98b; Rlf = 100% rice drink fermented with *Lb. fermentum* MR13; Rbb = 100% rice drink fermented with *B. bifidum* B700795; Rc = 100% rice drink fermented with bacterial consortium; Slp = 100% soy drink fermented with *Lb. plantarum* 98b; Slf = 100% soy drink fermented with *Lb. fermentum* MR13; Sbb = 100% soy drink fermented with *B. bifidum* B700795; Sc = 100% soy drink fermented with bacterial consortium; HRlp = 50% hemp seed drink and 50% rice drink fermented with *Lb. plantarum* 98b; HRlf = 50% hemp seed drink and 50% rice drink fermented with *Lb. fermentum* MR13; HRbb = 50% hemp seed drink and 50% rice drink fermented with *B. bifidum* B700795; HRc = 50% hemp seed drink and 50% rice drink fermented with bacterial consortium; HSlp = 50% hemp seed drink and 50% soy drink fermented with *Lb. plantarum* 98b; HSlf = 50% hemp seed drink and 50% soy drink fermented with *Lb. fermentum* MR13; HSbb = 50% hemp seed drink and 50% soy drink fermented with *B. bifidum* B700795; HSc = 50% hemp seed drink and 50% soy drink fermented with bacterial consortium; SRlp = 50% soy drink and 50% rice drink fermented with *Lb. plantarum* 98b; SRlf = 50% soy drink and 50% rice drink fermented with *Lb. fermentum* MR13; SRbb = 50% soy drink and 50% rice drink fermented with *B. bifidum* B700795; SRc = 50% soy drink and 50% rice drink fermented with bacterial consortium.

**Figure 3 microorganisms-07-00509-f003:**
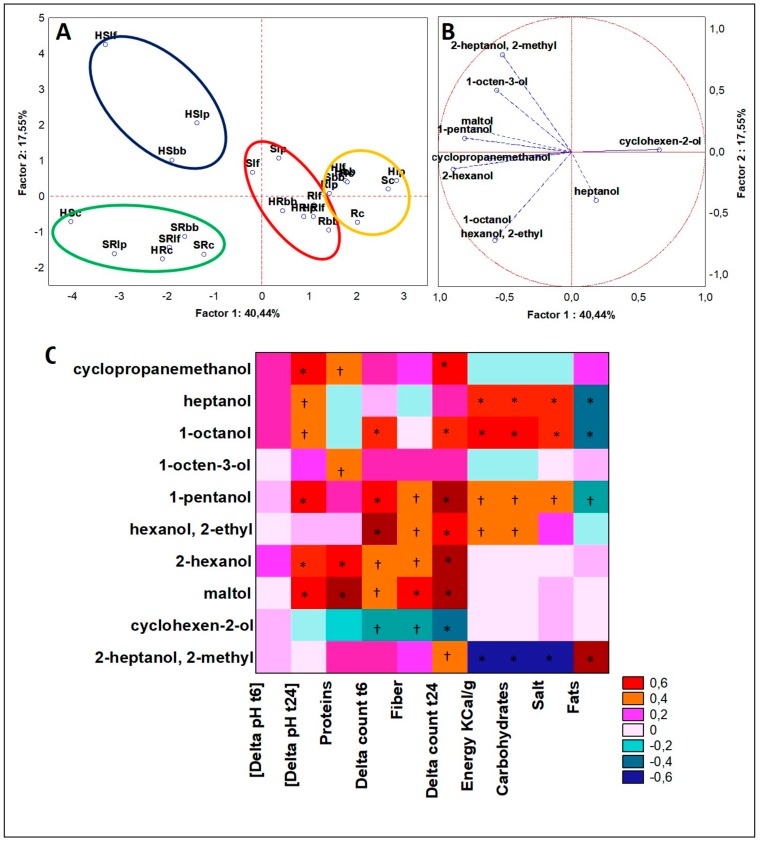
(**A**) Principal Component Analysis (PCA) of cases and (**B**) variables on alcohols; (**C**) Spearman rank correlations on alcohols. * *p* < 0.01; † *p* < 0.05. For samples abbreviations see Figure 1. Figure 3A: cluster 1 = blue line; cluster 2 = red line; cluster 3 = green line; cluster 4 = yellow line.

**Figure 4 microorganisms-07-00509-f004:**
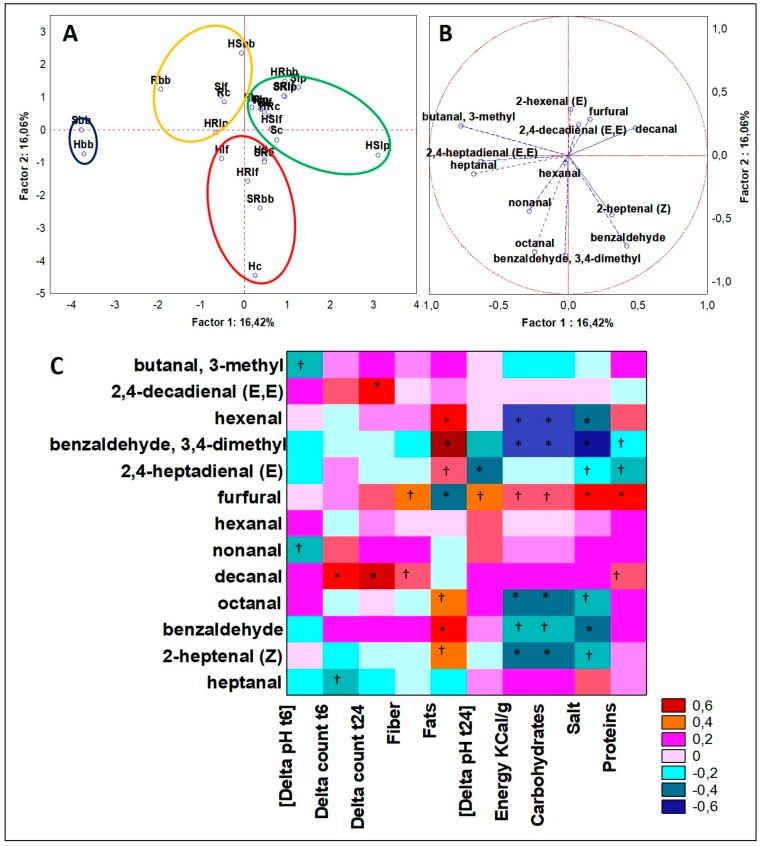
(**A**) PCA of cases and (**B**) variables on aldehydes; (**C**) Spearman rank correlations on aldehydes. * *p* < 0.01; † *p* < 0.05. For samples abbreviations see Figure 1. Figure 4a: cluster 1 = blue line; cluster 2 = red line; cluster 3 = green line; cluster 4 = yellow line.

**Figure 5 microorganisms-07-00509-f005:**
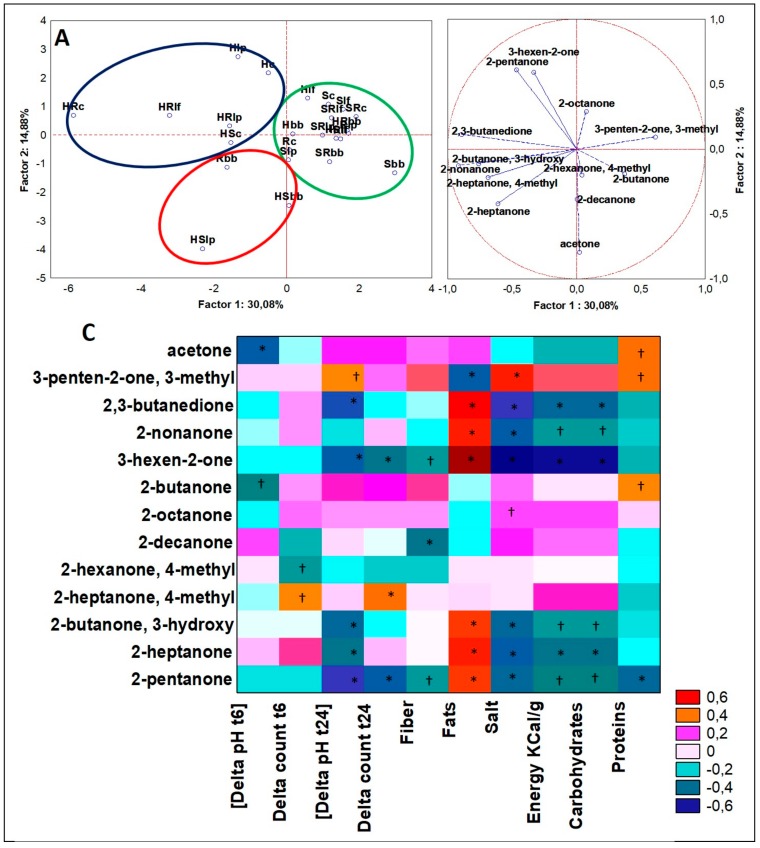
(**A**) PCA of cases and (**B**) variables on ketones; **c)** Spearman rank correlations on ketones. * *p* < 0.01; † *p* < 0.05. For samples abbreviations see Figure 1. Figure 5A: cluster 1 = blue line; cluster 2 = red line; cluster 3 = green line.3.4.4. Alkanes.

**Figure 6 microorganisms-07-00509-f006:**
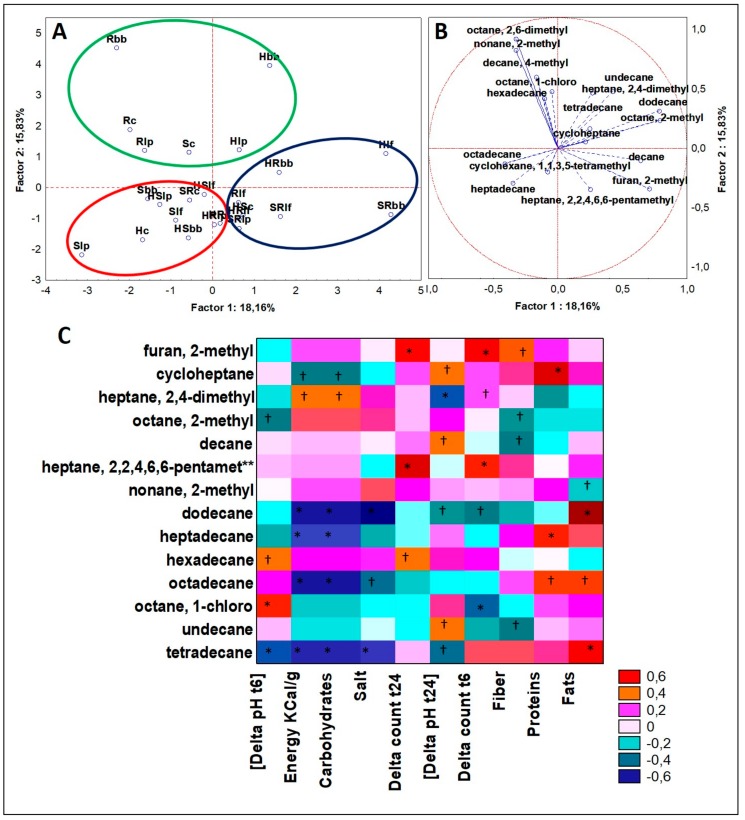
(**A**) PCA of cases and (**B**) variables on alkanes; (**C**) Spearman rank correlations on alkanes. * *p* < 0.01; † *p* < 0.05. **heptane,2,2,4,6,6-pentamethyl. For samples abbreviations see Figure 1. Figure 6A: cluster 1 = blue line; cluster 2 = red line; cluster 3 = green line.3.4.5. Organic Acids.

**Figure 7 microorganisms-07-00509-f007:**
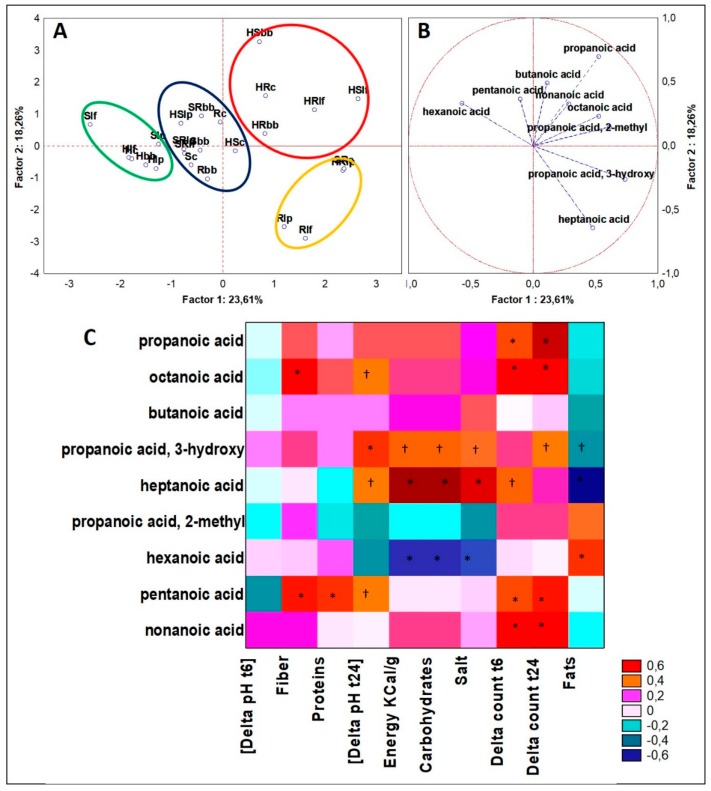
(**A**) PCA of cases and (**B**) variables on fatty acids; (**C**) Spearman rank correlations on fatty acids. * *p* < 0.01; † *p* < 0.05. For samples abbreviations see Figure 1. Figure 7A: cluster 1 = blue line; cluster 2 = red line; cluster 3 = green line; cluster 4 = yellow line.

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
