# Peer review of "Shift of Aromatic Profile in Probiotic Hemp Drink Formulations: A Metabolomic Approach"

_microorganisms, 2019, doi:10.3390/microorganisms7110509_

Round 1

Reviewer 1 Report

[Microbial loads ... and controlled afterwards ...] it should ... and measured afterwards

Please check the wording across the text; there are minor points that should be checked carefully 

Author Response

1) [Microbial loads ... and controlled afterwards ...] it should ... and measured afterwards

LN) changed accordingly

2) Please check the wording across the text; there are minor points that should be checked carefully

LN) minor points, as punctuation and wording has been revised

Reviewer 2 Report

General comment: The research article entitled “Shift of aromatic profile in probiotic hemp drink formulations: a metabolomic approach” is a well-organized study, with sufficient methodology and adequate description of the results. Some minor corrections are required for the improvement of the manuscript.

Abstract: The Abstract is well written and adequately presents the aim and the basic results of the study.

-Line 16. Authors could add a short sentence about the methods used.

Introduction: The introduction section is well-written and adequately covers the importance of finding of alternative drinks such as hemp drink.

-Line 32. “As reviewed by 2” could be replaced by “as recently has reviewed” or something else.

-Line 26. New paragraph should be started.

Materials and Methods:  The materials and methods are adequately presented. 

Results: The results of the study are analytically presented. Tables and Figures are adequate explain the findings of the study.

Discussion: The results of study are sufficiently discussed.

-Discussion session should be divided to different paragraphs.

Conclusion: The conclusion is adequate and summarizes the main text.

Bibliography/References: The references used by the authors cover adequately the relative scientific field and the aims of the study

Author Response

1) Line 16. Authors could add a short sentence about the methods used.

LN) This sentence was added: The drinks were characterized for their microbial growth, by means of culture-dependent and independent techniques, and for their volatilome, by means of SPME-GC-MS analysis.

2) Line 32. “As reviewed by 2” could be replaced by “as recently has reviewed” or something else.

LN) the sentence as recently has reviewed was added

3) Line 26. New paragraph should be started.

LN) done accordingly

4) Discussion session should be divided to different paragraphs.

LN) done accordingly

Reviewer 3 Report

The authors have proposed to use a metabolomic analysis of aroma volatile substances for estimation of process microbial probiotic fermentation of hemp seed milk and compositions thereof with soya bean and rice milk. They have used solid phase microextraction-gas chromatography-mass spectrometry (SPIME-GC-MS) and necessary statistic procedure in order to find any correlations between contents of different aldehydes, ketons, alkans, alcohols, fatty acids and salts, energy, carbohydrates stage of fermentation, probiotic strains and blend formulation. The showed that a specific aroma and its compounds of the hemp seed milk beverages are dependent to the type of formulation and to the probiotic used. In particular they have found that 2-heptanol, 2-methyl, 2,4-decadienal, 2-butanone, 3-hydroxy, 2,4-butanedione, and propanoic acid were good descriptors of probiotics fermentations in hemp seed drinks. The proposed analytic procedure may be useful for control of probiotic fermentation of the hemp seed milk and corresponding blends.

The article is clearly written, the introductive part is very useful for understanding of the goals of the investigation. The article may be interesting for readers.

However, the description of main experimental procedure – SPME-GC-MS – is absolutely absent. I suppose, that a brief description of the analytic procedure is necessary.

Hence the article may be published after minor revision.

Author Response

1) However, the description of main experimental procedure – SPME-GC-MS – is absolutely absent. I suppose, that a brief description of the analytic procedure is necessary.

LN) A more detailed description was provided:

Before each head space sampling, the fiber was exposed to the GC inlet for 10 min for thermal desorption at 250 °C in a blank sample. Five grams of dough or bread were minced and placed in 10 mL vials and sealed. The samples were then equilibrated for 10 min at 50 °C. The SPME fiber was exposed to each sample for 40 min and finally the fiber was inserted into the injection port of the GC for a 10 min sample desorption. The temperature program was: 50 °C for 0 min then programmed at 1.5 °C/min to 65 °C and finally at 3.5 °C/min to 220 °C, which was maintained for 20 min. Injector, interface and ion source temperatures were 250, 250 and 230 °C, respectively. Injections were carried out in splitless mode and helium (3 mL/min) was used as carrier gas